# ReSSL: Relational Self-Supervised Learning with Weak Augmentation

**Mingkai Zheng**[1,2]    **Shan You**[2,4*]    **Fei Wang**[3]
**Chen Qian**[2]    **Changshui Zhang**[4]    **Xiaogang Wang**[2,5]    **Chang Xu**[1]

[1]School of Computer Science, Faculty of Engineering, The University of Sydney
[2]SenseTime Research    [3]University of Science and Technology of China
[4]Department of Automation, Tsinghua University,
Institute for Artificial Intelligence, Tsinghua University (THUAI),
Beijing National Research Center for Information Science and Technology (BNRist)
[5]The Chinese University of Hong Kong

## Abstract

Self-supervised Learning (SSL) including the mainstream contrastive learning has achieved great success in learning visual representations without data annotations. However, most of methods mainly focus on the instance level information (*i.e.*, the different augmented images of the same instance should have the same feature or cluster into the same class), but there is a lack of attention on the relationships between different instances. In this paper, we introduced a novel SSL paradigm, which we term as relational self-supervised learning (ReSSL) framework that learns representations by modeling the relationship between different instances. Specifically, our proposed method employs sharpened distribution of pairwise similarities among different instances as *relation* metric, which is thus utilized to match the feature embeddings of different augmentations. Moreover, to boost the performance, we argue that weak augmentations matter to represent a more reliable relation, and leverage momentum strategy for practical efficiency. Experimental results show that our proposed ReSSL significantly outperforms the previous state-of-the-art algorithms in terms of both performance and training efficiency. Code is available at https://github.com/KyleZheng1997/ReSSL

## 1  Introduction

Recently, self-supervised learning (SSL) has shown its superiority and achieved promising results for unsupervised visual representation learning in computer vision tasks [40, 27, 32, 6, 9, 47, 23, 24]. The purpose of a typical self-supervised learning algorithm is to learn general visual representations from a large amount of data without human annotations, which can be transferred or leveraged in downstream tasks (*e.g.*, classification, detection, and segmentation). Some previous works [5, 23] even have proven that a good unsupervised pretraining can lead to a better downstream performance than supervised pretraining.

Among various SSL algorithms, contrastive learning [47, 45, 6] serves as a state-of-the-art framework, which mainly focuses on learning an invariant feature from different views. For example, instance discrimination is a widely adopted pre-text task as in [6, 24, 47], which utilizes the noisy contrastive estimation (NCE) to encourage two augmented views of the same image to be pulled closer on the embedding space but pushes apart all the other images away. Deep Clustering [4, 48, 5] is an alternative pre-text task that forces different augmented views of the same instance to be clustered into the same class. However, instance discrimination based methods will inevitably induce a class

---

[*]Corresponding author `youshan@sensetime.com`

35th Conference on Neural Information Processing Systems (NeurIPS 2021).

collision problem [1, 36, 10], where similar images should be pulled closer instead of being pushed away. Deep clustering based methods cooperated with traditional clustering algorithms to assign a label for each instance, which relaxed the constraint of instance discrimination, but most of these algorithms adopt a strong assumption, *i.e.*, the labels must induce an equipartition of the data, which might introduce some noise and hurt the learned representations.

In this paper, we introduce a novel Relational Self-Supervised Learning framework (ReSSL), which does not encourage explicitly to push away different instances, but uses *relation* as a manner to investigate the inter-instance relationships and highlight the intra-instance invariance. Concretely, we aim to maintain the consistency of pairwise similarities among different instances for two different augmentations. For example, if we have three instances $x^1$, $x^2$, $y$ and $z$ where $x^1$, $x^2$ are two different augmentations of $x$, $y$ and $z$ are different samples. Then, if $x^1$ is similar to $y$ but different to $z$, we wish $x^2$ can maintain such relationship and vice versa. In this way, the relation can be modelled as a similarity distribution between a set of augmented images, and then use it as a metric to align the same images with different augmentations, so that the relationship between different instances could be maintained across different views.

However, this simple manner induces unexpectedly horrible performance if we follow the same training recipe as other contrastive learning methods [6, 24]. We argue that construction of a proper relation matters for ReSSL; aggressive data augmentations as in [6, 7, 41] are usually leveraged by default to generate diverse positive pairs that increase the difficulty of the pre-text task. However, this hurts the reliability of the target relation. Views generated by aggressive augmentations might cause the loss of semantic information, so the target relation might be noisy and not that reliable. In this way, we propose to leverage weaker augmentations to represent the relation, since much lesser disturbances provide more stable and meaningful relationships between different instances. Besides, we also sharpen the target distribution to emphasize the most important relationship and utilize the memory buffer with a momentum-updated network to reduce the demand of large batch size for more efficiency. Experimental results on multiple benchmark datasets show the superiority of ReSSL in terms of both performance and efficiency. For example, with 200 epochs of pre-training, our ReSSL achieved 69.9% on ImageNet [14] linear evaluation protocol, which is 2.4% higher than our baseline method (MoCoV2 [8]). When working with the Multi-Crop strategy (200 epochs), ReSSL achieved new state-of-the-art 74.7% Top-1 accuracy, which is 1.4% higher than CLSA-Multi [46].

Our contributions can be summarized as follows.

- We proposed a novel SSL paradigm, which we term it as relational self-supervised learning (ReSSL). ReSSL maintains the relational consistency between the instances under different augmentations instead of explicitly pushing different instances away.

- Our proposed weak augmentation and sharpening distribution strategy provide a stable and high quality target similarity distribution, which makes the framework works well.

- ReSSL is a simple and effective SSL framework since it replaces the widely adopted contrastive loss with our proposed relational consistency loss. It achieved state-of-the-art performance under the same training cost.

## 2  Related Work

**Self-Supervised Learning**. Early works in self-supervised learning methods rely on all sorts of pretext to learn visual representations. For example, colorizing gray-scale images [50], image jigsaw puzzle [39], image super-resolution [34], image inpainting [19], predicting a relative offset for a pair of patches [16], predicting the rotation angle [35], and image reconstruction [2, 22, 3, 17]. Although these methods have shown their effectiveness, they lack the generality of the learned representations.

**Instance Discrimination**. The recent contrastive learning methods [32, 40, 6, 24, 41, 38, 29, 27, 30] have made a lot of progress in the field of self-supervised learning. Most of the previous contrastive learning methods are based on the instance discrimination [47] task in which positive pairs are defined as different views of the same image, while negative pairs are formed by sampling views from different images. SimCLR [6, 7] shows that image augmentation (*e.g.*Grayscale, Random Resized Cropping, Color Jittering, and Gaussian Blur), nonlinear projection head and large batch size plays a critical role in contrastive learning. Since large batch size usually requires a lot of GPU memory, which is not very friendly to most of researchers. MoCo [24, 8] proposed a momentum contrast

mechanism that forces the query encoder to learn the representation from a slowly progressing key encoder and maintain a memory buffer to store a large number of negative samples. InfoMin [41] proposed a set of stronger augmentation that reduces the mutual information between views while keeping task-relevant information intact. AlignUniform [45] shows that alignment and uniformity are two critical properties of contrastive learning.

**Deep Clustering**. In contrast to instance discrimination which treats every instance as a distinct class, deep clustering [4] adopts the traditional clustering method (*e.g.*KMeans) to label each image iteratively. Eventually, similar samples will be clustered into the same class. Simply apply the KMeans algorithm might lead to a degenerate solution where all data points are mapped to the same cluster; SeLa [48] solved this issue by adding the constraint that the labels must induce equipartition of the data and proposed a fast version of the Sinkhorn-Knopp to achieve this. SwAV [5] further extended this idea and proposed a scalable online clustering framework. PCL [36] reveals the class collision problem and simply performed instance discrimination and unsupervised clustering simultaneously; although it gets the same linear classification accuracy with MoCoV2, it has better performance on downstream tasks.

**Contrastive Learning Without Negatives**. Most previous contrastive learning methods prevent the model collapse in an explicit manner (*e.g.* push different instances away from each other or force different instances to be clustered into different groups.) BYOL [23] can learn a high-quality representation without negatives. Specifically, it trains an online network to predict the target network representation of the same image under a different augmented view and using an additional predictor network on top of the online encoder to avoiding the model collapse. SimSiam [9] shows that simple Siamese networks can learn meaningful representations even without the use of negative pairs, large batch size, and momentum encoders.

## 3 Methodology

In this section, we will first revisit the preliminary work on contrastive learning; then, we will introduce our proposed relational self-supervised learning framework. After that, the algorithm and the implementation details will also be explained.

### 3.1 Preliminaries on Self-supervised Learning

Given $N$ unlabeled samples $\mathbf{x}$, we randomly apply a composition of augmentation functions $T(\cdot)$ to obtain two different views $\mathbf{x}^1$ and $\mathbf{x}^2$ through $T(\mathbf{x}, \theta_1)$ and $T(\mathbf{x}, \theta_2)$ where $\theta$ is the random seed for $T$. Then, a convolutional neural network based encoder $\mathcal{F}(\cdot)$ is employed to extract the information from these samples, i.e., $\mathbf{h} = \mathcal{F}(T(\mathbf{x}, \theta))$. Finally, a two-layer non-linear projection head $g(\cdot)$ is utilized to map $\mathbf{h}$ into embedding space, which can be written as: $\mathbf{z} = g(\mathbf{h})$. SimCLR [6] and MoCo [24] style framework adopt the noise contrastive estimation (NCE) objective for discriminating different instances in the dataset. Suppose $\mathbf{z}_i^1$ and $\mathbf{z}_i^2$ are the representations of two augmented views of $\mathbf{x}_i$ and $\mathbf{z}_k$ is a different instance. The NCE objective can be expressed by Eq. (1), where the similarity function $sim(\cdot)$ represents the dot product between $L_2$ normalized vectors $sim(\mathbf{u}, \mathbf{v}) = \mathbf{u}^T\mathbf{v}/\|\mathbf{u}\|\|\mathbf{v}\|$ and $\tau$ is the temperature parameter.

$$\mathcal{L}_{NCE} = -\log \frac{\exp(sim(\mathbf{z}^1, \mathbf{z}^2)/\tau)}{\exp(sim(\mathbf{z}_i^1, \mathbf{z}_i^2)/\tau) + \sum_{k=1}^{N} \exp(sim(\mathbf{z}_i^1, \mathbf{z}_k)/\tau)}. \tag{1}$$

BYOL [23] and SimSiam [9] style framework add an additional non-linear predictor head $q(\cdot)$ which further maps $\mathbf{z}$ to $\mathbf{p}$. The model will minimize the negative cosine similarity (equivalent to minimize the L2 distance) between $\mathbf{z}$ to $\mathbf{p}$.

$$\mathcal{L}_{cos} = -\frac{\mathbf{p}^1}{\|\mathbf{p}^1\|} \cdot \frac{\mathbf{z}^2}{\|\mathbf{z}^2\|}, \qquad\qquad \mathcal{L}_{mse} = \|\mathbf{p}^1 - \mathbf{z}^2\|_2^2. \tag{2}$$

Tricks like stop-gradient and momentum teacher are often applied to avoid model collapsing.

### 3.2 Relational Self-Supervised Learning

In classical self-supervised learning, different instances are to be pushed away from each other, and augmented views of the same instance is expected to be of exactly the same features. However, both

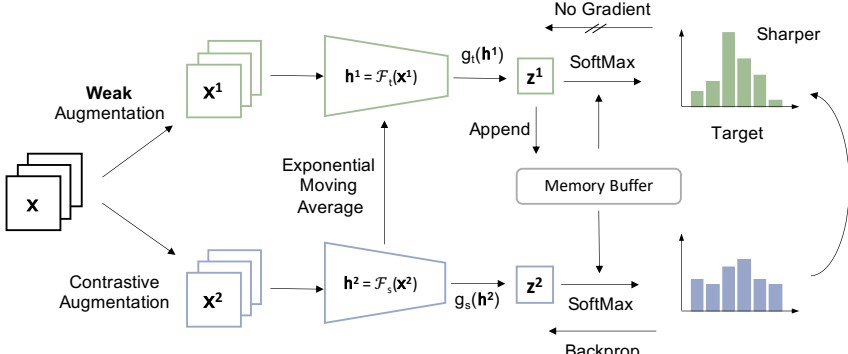

Figure 1: The overall framework of our proposed method. We adopt the student-teacher framework where the student is trained to predict the representation of the teacher, and the teacher is updated with a "momentum update" (exponential moving average) of the student. The relationship consistency is achieve by align the conditional distribution for student and teacher model. Please see more details in our method part.

constrains are too restricted because of the existence of similar samples and the distorted semantic information if aggressive augmentation is adopted. In this way, we do not encourage explicit negative instances (those to be pushed away) for each instance; instead, we leverage the pairwise similarities as a manner to explore their relationships. And we pull the features of two different augmentations in this sense of relation metric. As a result, our method relaxes both (1) and (2), where different instances do not always need to be pushed away from each other; and augmented views of the same instance only need to share the similar but not exactly the same features.

Concretely, given a image $\mathbf{x}$ in a batch of samples , two different augmented views can be obtained by $\mathbf{x}^1 = T(\mathbf{x}, \theta_1)$, $\mathbf{x}^2 = T(\mathbf{x}, \theta_2)$ and calculate the corresponds embedding $\mathbf{z}^1 = g(\mathcal{F}(\mathbf{x}^1))$, $\mathbf{z}^2 = g(\mathcal{F}(\mathbf{x}^2))$. Then, we calculate the similarities between the instances of the first augmented images. Which can be measured by $sim(\mathbf{z}^1, \mathbf{z}_i)$. A softmax layer can be adopted to process the calculated similarities, which then produces a relationship distribution:

$$\mathbf{p}_i^1 = \frac{\exp(sim(\mathbf{z}^1, \mathbf{z}_i)/\tau_t)}{\sum_{k=1}^K \exp(sim(\mathbf{z}^1, \mathbf{z}_k)/\tau_t,)}. \tag{3}$$

where $\tau_t$ is the temperature parameter. At the same time, we can calculate the relationship between $\mathbf{x}^2$ and the $i$-th instance as $sim(\mathbf{z}^2, \mathbf{z}_i)$. The resulting relationship distribution can be written as:

$$\mathbf{p}_i^2 = \frac{\exp(sim(\mathbf{z}^2, \mathbf{z}_i)/\tau_s)}{\sum_{k=1}^K \exp(sim(\mathbf{z}^2, \mathbf{z}_k)/\tau_s,)}. \tag{4}$$

where $\tau_s$ is a different temperature parameter. We propose to push the relational consistency between $p_i^1$ and $p_i^2$ by minimizing the Kullback–Leibler divergence, which can be formulated as:

$$\mathcal{L}_{relation} = D_{KL}(\mathbf{p}^1 || \mathbf{p}^2) = H(\mathbf{p}^1, \mathbf{p}^2) - H(\mathbf{p}^1). \tag{5}$$

Since the $\mathbf{p}^1$ will only be used as a target, we only minimize $H(\mathbf{p}^1, \mathbf{p}^2)$ in our implementation.

**More efficiency with Momentum targets.** However, the quality of the target similarity distribution $\mathbf{p}^1$ is crucial, to make the similarity distribution reliable and stable, we usually require a large batch size which is very unfriendly to GPU memories. To resolve this issue, we utilize a "momentum update" network as in [24, 8], and maintain a large memory buffer $\mathcal{Q}$ of $K$ past samples $\{\mathbf{z}_k | k = 1, ..., K\}$ (following the FIFO principle) for storing the feature embeddings from the past batches, which can then be used for simulating the large batch size relationship and providing a stable similarity distribution.

$$\mathcal{F}_t \leftarrow m\mathcal{F}_t + (1 - m)\mathcal{F}_s, \quad g_t \leftarrow mg_t + (1 - m)g_s, \tag{6}$$

where $\mathcal{F}_s$ and $g_s$ denote the most latest encoder and head, respectively, so we name them as the student model with a subscript $s$. On the other hand, $\mathcal{F}_t$ and $g_t$ stand for ensembles of the past encoder and head, respectively, so we name them as the teacher model with a subscript $t$. $m$ represents the momentum coefficient which controls how fast the teacher $\mathcal{F}_t$ will be updated.

**Sharper Distribution as Target**. Note, the value of $\tau_t$ has to be smaller than $\tau_s$ since $\tau_t$ will be used to generate the target distribution. A smaller $\tau$ will result in a "sharper" distribution which can be

interpreted as highlight the most similar feature for $\mathbf{z}^1$. Align $\mathbf{p}^2$ with $\mathbf{p}^1$ can be regarded as pulling $\mathbf{z}^2$ towards the features that are similar with $\mathbf{z}^1$.

**Weak Augmentation Strategy for Teacher**. To further improve the quality and stability of the target distribution, we adopt a weak augmentation strategy for the teacher model since the standard contrastive augmentation is too aggressive, which introduced too many disturbances and will mislead the student network. Please refer to more details in our empirical study.

**Compare with SEED and CLSA**. SEED [21] follows the standard Knowledge Distillation (KD) paradigm [26, 49, 18] where it aims to distill the knowledge from a larger network into a smaller architecture. The knowledge transfer happens in the same view but between different models. In our framework, we are trying to maintain the relational consistency between different augmentations; the knowledge transfer happens between different views but in the same network. CLSA [46] also introduced the concept of using weak augmentation to guide a stronger augmentation. However, the "weak" augmentation in CLSA is equivalent to the "strong" augmentation in our method (We do not use any stronger augmentations such as [12, 13]). On the other hand, CLSA still adopts the InfoNCE loss (1) for instance discrimination, where our proposed method only utilized the relational consistency loss (5). Finally, CLSA requires at least one additional sample during training, which will slow down the training speed.

---

**Algorithm 1:** Relational Self-supervised Learning with Weak Augmentation (ReSSL)

---

**Input :** $\mathbf{x}$: a batch of samples. $T_w(\cdot)$: Weak augmentation function. $T_c(\cdot)$: Contrastive augmentation function. $\mathcal{F}_t$ and $\mathcal{F}_s$: the teacher and student backbone network. $g_t$ and $g_s$: the non-linear projection head for teacher and student. $Q$: the memory buffer

**while** *network not converge* **do**
    **for** *i=1 to step* **do**
        Fetch $\mathbf{x}$ from current batch $\mathcal{B}$
        $\mathbf{z}^1 = g_t(\mathcal{F}_t(T_w(\mathbf{x}, \theta_1)));$      $\mathbf{z}^2 = g_s(\mathcal{F}_s(T_c(\mathbf{x}, \theta_2)));$
        $\mathbf{p}^1 = \text{SoftMax}(\mathbf{z}^1 Q^T / \tau_t);$      $\mathbf{p}^2 = \text{SoftMax}(\mathbf{z}^2 Q^T / \tau_s);$       `// Eq.` (3)(4)
        Calculate $\mathcal{L}_{relation}$ loss by CrossEntropy($\mathbf{p}^1, \mathbf{p}^2$);       `// Eq.` (5)
        Update $\mathcal{F}_s$ and $g_s$ with loss $\mathcal{L}_{relation}$
        Update $\mathcal{F}_t$ and $g_t$ by $\mathcal{F}_t \leftarrow m\mathcal{F}_t + (1-m)\mathcal{F}_s, g_t \leftarrow mg_t + (1-m)g_s$; `// Eq.` (6)
        Update the memory buffer $Q$ by $\mathbf{z}^1$
    **end**
**end**
**Output :** The well trained model $\mathcal{F}_s$

---

## 4   Empirical Study

In this section, we will empirically study our proposed method on 4 popular self-supervised learning benchmarks and compare to previous state-of-the-art algorithms (SimCLR [6], BYOL [23], SimSiam [9], MoCoV2 [8]).

**Small Dataset**. CIFAR-10 and CIFAR-100 [31]. The CIFAR-10 dataset consists of 60000 32x32 colour images in 10 classes, with 6000 images per class. There are 50000 training images and 10000 test images. CIFAR-100 is just like the CIFAR-10, except it has 100 classes containing 600 images each. There are 500 training images and 100 testing images per class.

**Medium Dataset**. STL-10 [11] and Tiny ImageNet [33]. STL10 [11] dataset is composed of 96x96 resolution images of 10 classes, 5K labeled training images, 8K validation images, and 100K unlabeled images. The Tiny ImageNet dataset is composed of 64x64 resolution images of 200 classes with 100K training images and 10k validation images.

**Implementation Details** We adopt the ResNet18 [25] as our backbone network. Because most of our dataset contains low-resolution images, we replace the first 7x7 Conv of stride 2 with 3x3 Conv of stride 1 and remove the first max pooling operation for a small dataset. For data augmentations, we use the random resized crops (the lower bound of random crop ratio is set to 0.2), color distortion (strength=0.5) with a probability of 0.8, and Gaussian blur with a probability of 0.5. The images from the small and medium datasets will be resized to 32x32 and 64x64 resolution respectively. Our method is based on MoCoV2 [8]; in order to simulate the shuffle BN trick on one GPU, we simply divide a batch of data into different groups and then calculate BN statistics within each group. The

Table 1: Compare to other SSL algorithms on small and medium dataset.

| Method | BackProp | EMA | CIFAR-10 | CIFAR-100 | STL-10 | Tiny ImageNet |
|---|---|---|---|---|---|---|
| Supervised | - | - | 94.22 | 74.66 | 82.55 | 59.26 |
| SimCLR [6] | 2x | No | 84.92 | 59.28 | 85.48 | 44.38 |
| BYOL [23] | 2x | Yes | 85.82 | 57.75 | 87.45 | 42.70 |
| SimSiam [9] | 2x | No | 88.51 | 60.00 | 87.47 | 37.04 |
| MoCoV2 [8] | 1x | Yes | 86.18 | 59.51 | 85.88 | 43.36 |
| ReSSL (Ours) | 1x | Yes | **90.20** | **63.79** | **88.25** | **46.60** |

momentum value and memory buffer size are set to 0.99/0.996 and 4096/16384 for small and medium datasets respectively. Moreover, The model is trained using SGD optimizer with a momentum of 0.9 and weight decay of $5e^{-4}$. We linear warm up the learning rate for 5 epochs until it reaches $0.06 \times BatchSize/256$, then switch to the cosine decay scheduler [37].

**Evaluation Protocol**. All the models will be trained for 200 epochs. For testing the representation quality, we evaluate the pre-trained model on the widely adopted linear evaluation protocol - We will freeze the encoder parameters and train a linear classifier on top of the average pooling features for 100 epochs. To test the classifier, we use the center crop of the test set and computes accuracy according to predicted output. We train the classifier with a learning rate of 30, no weight decay, and momentum of 0.9. The learning rate will be times 0.1 in 60 and 80 epochs. Note, for STL-10; the pretraining will be applied on both labeled and unlabeled images. During the linear evaluation, only the labeled 5K images will be used.

**Result**. As we can see the result in Table 1, our proposed method outperforms the previous method on all four benchmarks. Reminder, most of the previous method requires twice back-propagation, which results in a much higher training cost than MoCoV2 and our method.

## 4.1 A Properly Sharpened Relation is A Better Target

The temperature parameter is very crucial in most contrastive learning algorithms. To verify the effective of $\tau_s$ and $\tau_t$ for our proposed method, we fixed $\tau_s = 0.1$ or $0.2$, and sweep over $\tau_t = \{0.01, 0.02, ..., 0.07\}$. The result is shown in Table 2. For $\tau_t$, the optimal value is either $0.04$ or $0.05$ across all different datasets. As we can see, the performance is increasing when we increase $\tau_t$ from 0 to 0.04 and 0.05. After that, the performance will start to decrease. Note, $\tau_t \to 0$ correspond to the Top-1 or $argmax$ operation which produce a one-hot distribution as the target. On the other hand, when $\tau_t \to 0.1$, the target will be a much flatter distribution that cannot highlight the most similar features for students. Hence, $\tau_t$ can not be either too small or too large, but it has to be smaller than $\tau_s$ ($\mathbf{p}^1$ has to be sharper than $\mathbf{p}^2$) so that the target distribution can provide effective guidance to the student model.

Table 2: Effect of different $\tau_t$ and $\tau_s$ for ReSSL

| Dataset | $\tau_s$ | $\tau_t = 0.01$ | $\tau_t = 0.02$ | $\tau_t = 0.03$ | $\tau_t = 0.04$ | $\tau_t = 0.05$ | $\tau_t = 0.06$ | $\tau_t = 0.07$ |
|---|---|---|---|---|---|---|---|---|
| CIFAR-10 | 0.1 | 89.35 | 89.74 | 90.09 | 90.04 | **90.20** | 90.18 | 88.67 |
| CIFAR-10 | 0.2 | 89.52 | 89.67 | 89.24 | 89.50 | 89.22 | 89.40 | 89.50 |
| CIFAR-100 | 0.1 | 62.34 | 62.79 | 62.71 | **63.79** | 63.46 | 63.20 | 61.31 |
| CIFAR-100 | 0.2 | 60.37 | 60.05 | 60.24 | 60.09 | 59.09 | 59.12 | 59.76 |
| STL-10 | 0.1 | 86.65 | 86.96 | 87.16 | 87.32 | **88.25** | 87.83 | 87.08 |
| STL-10 | 0.2 | 85.17 | 86.12 | 85.01 | 85.67 | 85.21 | 85.51 | 85.28 |
| Tiny ImageNet | 0.1 | 45.20 | 45.40 | 46.30 | **46.60** | 45.08 | 45.24 | 44.18 |
| Tiny ImageNet | 0.2 | 43.28 | 42.98 | 43.58 | 42.12 | 42.70 | 42.76 | 42.60 |

For $\tau_t$, it is clearly to see that the result of $\tau_s = 0.1$ can always result a much higher performance than $\tau_s = 0.2$, which is different to MoCoV2 where $\tau_s = 0.2$ is the optimal value. According to [43, 44, 15], a greater temperature will result in a larger angular margin in the hypersphere. Since MoCoV2 adopts instance discrimination as the pretext task, a large temperature can enhance the compactness for the same instance and discrepancy for different instances. In contrast to instance discrimination, our method can be interpreted as pulling similar instances closer on the hypersphere; when the ground truth label is not available, the large angular margin might hurt the performance.

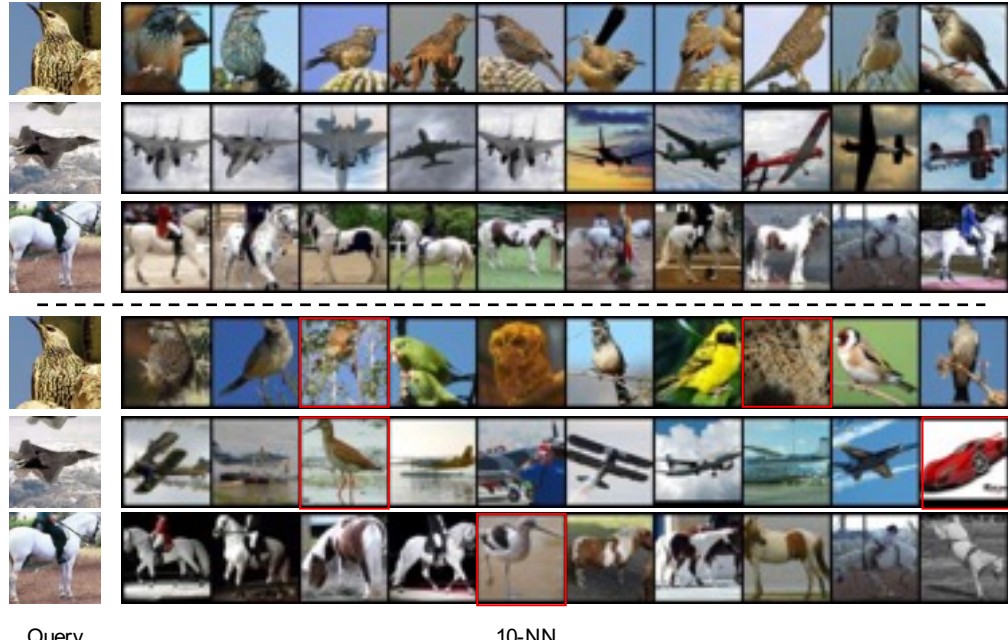

Query                                                      10-NN

Figure 2: Visualization of the 10 nearest neighbour of the query image. The top half is the result when we apply the weak augmentation. The bottom half is the case when the typical contrastive augmentation is adopted. Note, we use the red square to highlight the images that has different ground truth label with the query image.

## 4.2 Weak Augmentation Makes Better Relation

As we have mentioned, the weaker augmentation strategy for the teacher model is the key to the success of our framework. Here, We implement the weak augmentation as a random resized crop (the random ratio is set to $(0.2, 1)$) and a random horizontal flip. For temperature parameter, we simply adopt the same setting as in Table 2 and report the performance of the best setting. The result is shown in Table 3, as we can see that when we use the weak augmentation for the teacher model, the performance is significantly boosted across all datasets. We believe that this phenomenon is because relatively small disturbances in the teacher model can provide more accurate similarity guidance to the student model. To further verify this hypothesis, we random sampled three image from STL-10 training set as the query images, and then find the 10 nearest neighbour based on the weak / contrastive augmented query. We visualized the result in Figure 2,

Table 3: Effect of weak augmentation guided ReSSL

| Teacher Aug | Student Aug | CIFAR-10 | CIFAR-100 | STL-10 | Tiny ImageNet |
|---|---|---|---|---|---|
| Contrastive | Contrastive | 86.17 | 57.60 | 84.71 | 40.38 |
| Weak | Contrastive | **90.20** | **63.79** | **88.25** | **46.60** |

## 4.3 More Experiments on Weak Augmentation

Since the weak augmentation for the teacher model is one of the crucial points in ReSSL, we further analyze the effect of applying different augmentations on the teacher model. In this experiment, we simply set $\tau_t = 0.04$ and report the linear evaluation performance on the Tiny ImageNet dataset. The results are shown in Table 4. The first row is the baseline, where we simply resize all images to the same resolution (no extra augmentation is applied). Then, we applied random resized crops, random flip, color jitter, grayscale, gaussian blur, and various combinations. We empirically find that if we use no augmentation (*e.g.*, no random resized crops) for the teacher model, the performance tends to degrade. This might result from that the gap of features between two views is way too smaller, which undermines the learning of representations. However, too strong augmentations of teacher model will introduce too much noise and make the target distribution inaccurate (see Figure 2). Thus mildly weak augmentations are better option for the teacher, and random resized crops with random flip is the combination with the highest performance as Table 4 shows.

Table 4: Effect of different augmentation for teacher model (Tiny ImageNet)

| Random Resized Crops | Random Flip | Color Jitter | GrayScale | Gaussian Blur | Acc |
|:---:|:---:|:---:|:---:|:---:|:---:|
|  |  |  |  |  | 31.74 |
| ✓ |  |  |  |  | 46.00 |
|  | ✓ |  |  |  | 30.98 |
|  |  | ✓ |  |  | 29.46 |
|  |  |  | ✓ |  | 29.68 |
|  |  |  |  | ✓ | 30.10 |
| ✓ | ✓ |  |  |  | **46.60** |
| ✓ |  | ✓ |  |  | 44.44 |
| ✓ |  |  | ✓ |  | 42.28 |
| ✓ |  |  |  | ✓ | 44.88 |
| ✓ | ✓ | ✓ |  |  | 43.70 |
| ✓ | ✓ |  | ✓ |  | 42.28 |
| ✓ | ✓ |  |  | ✓ | 44.52 |

## 4.4 Dimension of the Relation

Since we also adopt the memory buffer as in MoCo [24], the buffer size will be equivalent to the dimension of the distribution $\mathbf{p}^1$ $\mathbf{p}^2$. Thus, it will be one of the crucial points in our framework. To verify the effect the memory buffer size, we simply keep $\tau_s = 0.1$ and $\tau_t = 0.04$, then varying the memory buffer size from 256 to 32768. The result is shown in Table 5, as we can see that a larger memory buffer can significantly boost the performance. However, a further increase in the buffer size can only bring a marginal improvement when the buffer is large enough.

Table 5: Effect of different memory buffer size on small and medium dataset

| Dataset (Small) | $K = 256$ | $K = 512$ | $K = 1024$ | $K = 4096$ | $K = 8192$ | $K = 16384$ |
|:---:|:---:|:---:|:---:|:---:|:---:|:---:|
| CIFAR-10 | 89.37 | 89.53 | 89.83 | 90.04 | 90.15 | 90.35 |
| CIFAR-100 | 61.17 | 62.47 | 63.20 | 63.79 | 63.84 | 64.06 |
| Dataset (Medium) | $K = 256$ | $K = 1024$ | $K = 4096$ | $K = 8192$ | $K = 16384$ | $K = 32768$ |
| STL-10 | 85.88 | 87.23 | 87.72 | 87.42 | 87.32 | 87.47 |
| Tiny ImageNet | 43.08 | 45.32 | 45.78 | 45.42 | 46.60 | 46.48 |

## 4.5 Visualization of Learned Representations

We also show the t-SNE [42] visualizations of the representations learned by our proposed method and MoCov2 on the training set of CIFAR-10. Our proposed relational consistency loss leads to better class separation than the contrastive loss.

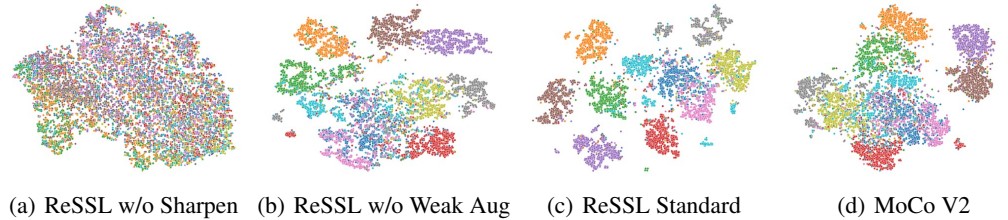

(a) ReSSL w/o Sharpen    (b) ReSSL w/o Weak Aug    (c) ReSSL Standard    (d) MoCo V2

Figure 3: t-SNE visualizations on CIFAR-10. Classes are indicated by colors.

## 5 Results on Large-scale Datasets

We also performed our algorithm on the large-scale ImageNet-1k dataset [14]. In the experiments, we adopt a learning rate of $0.05 * BatchSize/256$, a memory buffer size of 130k, and a 2-layer non-linear projection head with a hidden dimension 4096 and output dimension 512. For $\tau_t$ and $\tau_s$, we simply adopt the best setting from Table 2 where $\tau_t = 0.04$ and $\tau_s = 0.1$.

**Linear Evaluation**. For the linear evaluation of ImageNet-1k, we strictly follow the setting in SwAV [5]. The results are shown in Table 6. As we can see clearly that ReSSL consistently outperforms previous methods on both 1x and 2x backprop setting. (Please noted that the student network will be passed in one 224x224 augmented view and two 224x224 augmented views for 1x backprob and 2x backprob setting respectively.)

Table 6: Top-1 accuracy under the linear evaluation on ImageNet with the ResNet-50 backbone. The table compares the methods over 200 epochs of pretraining.

| Method | Arch | Backprop | EMA | Batch Size | Param | Epochs | Top-1 |
|---|---|---|---|---|---|---|---|
| Supervised | R50 | 1x | No | 256 | 24 | 120 | 76.5 |
| *1x Backprop Methods* | | | | | | | |
| InstDisc [47] | R50 | 1x | No | 256 | 24 | 200 | 58.5 |
| LocalAgg [52] | R50 | 1x | No | 128 | 24 | 200 | 58.8 |
| MoCo v2 [8] | R50 | 1x | Yes | 256 | 24 | 200 | 67.5 |
| MoCHi [30] | R50 | 1x | Yes | 512 | 24 | 200 | 68.0 |
| CPC v2 [32] | R50 | 1x | No | 512 | 24 | 200 | 63.8 |
| PCL v2 [36] | R50 | 1x | Yes | 256 | 24 | 200 | 67.6 |
| AdCo [28] | R50 | 1x | Yes | 256 | 24 | 200 | 68.6 |
| ReSSL (Ours) | R50 | 1x | Yes | 256 | 24 | 200 | **69.9** |
| *2x Backprop Methods* | | | | | | | |
| CLSA-Single [46] | R50 | 2x | Yes | 256 | 24 | 200 | 69.4 |
| SimCLR [6] | R50 | 2x | No | 4096 | 24 | 200 | 66.8 |
| SwAV [5] | R50 | 2x | No | 4096 | 24 | 200 | 69.1 |
| SimSiam [23] | R50 | 2x | No | 256 | 24 | 200 | 70.0 |
| BYOL [23] | R50 | 2x | Yes | 4096 | 24 | 200 | 70.6 |
| WCL [51] | R50 | 2x | No | 4096 | 24 | 200 | 70.3 |
| ReSSL (Ours) | R50 | 2x | Yes | 256 | 24 | 200 | **71.4** |

**Working with Multi-Crop Strategy**. We also performed ReSSL with Multi-Crop strategy. The result is shown below in Table 7. Specifically, the result of 4 crops is trained with the resolution of $224 \times 224, 160 \times 160, 128 \times 128, 96 \times 96$. For the result of 5 crops, we add an additional $192 \times 192$ image which is exactly the same with AdCo [28]. As we can see, our proposed ReSSL is significantly better than previous state-of-the-art methods.

Table 7: Working with Multi-Crop Strategy (Linear Evaluation on ImageNet)

| Method | Arch | EMA | Batch Size | Param | Epochs | Top-1 |
|---|---|---|---|---|---|---|
| SwAV [5] | R50 | No | 256 | 24 | 200 | 72.7 |
| AdCo [28] | R50 | No | 256 | 24 | 200 | 73.2 |
| CLSA-Multi [46] | R50 | Yes | 256 | 24 | 200 | 73.3 |
| ReSSL (4 crops) | R50 | Yes | 256 | 24 | 200 | **73.8** |
| ReSSL (5 crops) | R50 | Yes | 256 | 24 | 200 | **74.7** |

**Working with Smaller Architecture**. We also applied our proposed method on the smaller architecture (ResNet-18). The result is shown in Table 8. Following the same training recipe of the ResNet-50 in above, our proposed method has a higher performance than SEED [21] without a larger pretrained teacher network.

Table 8: Experiments on ResNet-18 (Linear Evaluation on ImageNet)

| Method | Epochs | Student | Teacher | Acc |
|---|---|---|---|---|
| MoCo v2 | 200 | ResNet-18 | EMA | 52.2 |
| SEED | 200 | ResNet-18 | ResNet-50 (MoCoV2) | 57.6 |
| ReSSL (1x backprop) | 200 | ResNet-18 | EMA | **58.1** |

**Low-shot Classification**. We further evaluate the quality of the learned representations by transferring them to other datasets. Following [36], we perform linear classification on the PASCAL

VOC2007 dataset [20]. Specifically, we resize all images to 256 pixels along the shorter side and taking a 224 × 224 center crop. Then, we train a linear SVM on top of corresponding global average pooled final representations. To study the transferability of the representations in few-shot scenarios, we vary the number of labeled examples $K$ and report the mAP. Table 9 shows the comparison between our method with previous works. We report the average performance over 5 runs (except for k=full).It's clearly to see that our proposed method is consistently outperform MoCo v2 and PCL v2 across all different $K$.

Table 9: Transfer learning on low-shot image classification

| Method | Epochs | ImageNet | $K$=16 | $K$=32 | $K$=64 | Full |
|---|---|---|---|---|---|---|
| Random | - | - | 10.10 | 11.34 | 11.96 | 12.42 |
| Supervised | 90 | 76.1 | 82.26 | 84.00 | 85.13 | 87.27 |
| MoCo V2 [8] | 200 | 67.5 | 76.14 | 79.16 | 81.52 | 84.60 |
| PCL V2 [36] | 200 | 67.5 | 78.34 | 80.72 | 82.67 | 85.43 |
| ReSSL (1x backprob) | 200 | 69.9 | **79.17** | **81.96** | **83.81** | **86.31** |

**Semi-Supervised Learning**. Next, we evaluate the performance obtained when fine-tuning the model representation using a small subset of labeled data. In this experiments, we adopt our 5 crops pre-trained model. The result is shown in Table 10. Notably, with just 200 epochs of pre-training, ReSSL outperforms all previous methods.

Table 10: Semi-supervised Learning

| Method | Epochs | Linear Eval | 1% Labels | 10% Labels |
|---|---|---|---|---|
| SimCLR [6] | 1000 | 69.3 | 48.3 | 65.6 |
| BYOL [23] | 1000 | 74.3 | 53.2 | 68.6 |
| SwAV [5] | 800 | 75.3 | 53.9 | 70.2 |
| ReSSL (5 crops) | 200 | 74.7 | **57.9** | **70.4** |

# 6 Conclusion

In this work, we propose relational self-supervised learning (ReSSL), a new paradigm for unsupervised visual representation learning framework that maintains the relational consistency between instances under different augmentations. Our proposed ReSSL relaxes the typical constraints in contrastive learning where different instances do not always need to be pushed away on the embedding space, and the augmented views do not need to share exactly the same feature. An extensive empirical study shows the effect of each component in our framework. The experiments on large-scaled datasets demonstrate the efficiency and state-of-the-art performance for unsupervised representation learning.

# Broader Impact

This work provides a technical advancement in the field of unsupervised visual representation learning. An immediate application of this work is to give a pre-trained model for the tasks where the data annotation is very hard to collect (*e.g.*medical images and fine-grained images.) Moreover, the most significant advantage of ReSSL is that we do not need to train the model for a long time as the previous method (generally 800 or 1000 epochs), which will cause a lot of carbon dioxide emissions. We believe ReSSL is a more environment-friendly method since it can achieve a competitive performance with much lesser training costs.

# Acknowledgment

This work is funded by the National Key Research and Development Program of China (No. 2018AAA0100701) and the NSFC 61876095. Chang Xu was supported in part by the Australian Research Council under Projects DE180101438 and DP210101859. Shan You is supported by Beijing Postdoctoral Research Foundation.

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
