## A Implementation Details on Small and Medium Dataset

We adopt the same backbone and data augmentation for all methods as we already described in Section 4. For SimCLR and BYOL, we use the LARS optimizer with a momentum of 0.9 and weight decay of $1e-4$; the learning rate will be linearly warmed up for 5 epochs until it reaches $1.0 \times BatchSize/256$. For linear evaluation, we use a standard SGD optimizer with a momentum of 0.9, weight decay of 0, and a learning rate of $0.2 \times BatchSize/256$; the learning rate will be cosine decayed for 100 epochs. For SimSiam, the optimizer, learning rate, weight decay, and the linear evaluation details are the same as our MoCo and ReSSL implementation (as in Section 4).

## B More Experiments on Temperature

In this section, we add more experiments for different $\tau_t$ (an extension for Table 2). As we can see, when $\tau_t \to \tau_s$, the model is simply collapsed, which further verified that $\tau_t$ has to be properly sharpened. *Note, as we have mentioned in Table 2, the optimal value for $\tau_t$ is 0.04∼0.05.*

Table 8: More experiments for different $\tau_t$ (Top-1 accuracy on small and medium dataset)

| $\tau_s$ | $\tau_t$ | CIFAR-10 | CIFAR-100 | STL-10 | Tiny ImageNet |
|---|---|---|---|---|---|
| 0.1 | 0.08 | 10.00 | 1.00 | 83.05 | 39.38 |
| 0.1 | 0.09 | 10.00 | 1.00 | 10.00 | 0.50 |
| 0.1 | 0.10 | 10.00 | 1.00 | 10.00 | 0.50 |

## C More Experiments on Weak Augmentation

Since the weak augmentation for the teacher model is one of the crucial points in ReSSL, we further analyze the effect of applying different augmentations on the teacher model. In this experiment, we simply set $\tau_t = 0.04$ and report the linear evaluation performance on the Tiny ImageNet dataset. The results are shown in Table 9. The first row is the baseline, where we simply resize all images to the same resolution (no extra augmentation is applied). Then, we applied random resized crops, random flip, color jitter, grayscale, gaussian blur, and various combinations. We empirically find that if we use no augmentation (*e.g.*, no random resized crops) for the teacher model, the performance tends to degrade. This might result from that the gap of features between two views is way too smaller, which undermines the learning of representations. However, too strong augmentations of teacher model will introduce too much noise and make the target distribution inaccurate (see Figure 2). Thus mildly weak augmentations are better option for the teacher, and random resized crops with random flip is the combination with the highest performance as Table 9 shows.

Table 9: Effect of different augmentation for teacher model (Tiny ImageNet)

| Random Resized Crops | Random Flip | Color Jitter | GrayScale | Gaussian Blur | Acc |
|---|---|---|---|---|---|
| | | | | | 31.74 |
| ✓ | | | | | 46.00 |
| | ✓ | | | | 30.98 |
| | | ✓ | | | 29.46 |
| | | | ✓ | | 29.68 |
| | | | | ✓ | 30.10 |
| ✓ | ✓ | | | | **46.60** |
| ✓ | | ✓ | | | 44.44 |
| ✓ | | | ✓ | | 42.28 |
| ✓ | | | | ✓ | 44.88 |
| ✓ | ✓ | ✓ | | | 43.70 |
| ✓ | ✓ | | ✓ | | 42.28 |
| ✓ | ✓ | | | ✓ | 44.52 |

## D Working with Smaller Architecture

We also applied our proposed method on the smaller architecture (ResNet-18). The result is shown in Table 10. Following the setting of ReSSL*, our proposed method has a higher performance than SEED [21] without a larger pretrained network.

Table 10: Experiments on ResNet-18 (Linear Evaluation on ImageNet)

| Method | Epochs | Student | Teacher | Acc |
|--------|--------|---------|---------|-----|
| MoCoV2 | 200 | ResNet-18 | EMA | 52.2 |
| SEED | 200 | ResNet-18 | ResNet-50 (MoCoV2) | 57.6 |
| ReSSL* | 200 | ResNet-18 | EMA | **58.1** |

## E Further Comparison on ImageNet with Similar Training Cost

In this section, we further add the multi-crop strategy for matching the training cost with $2\times$ backbprop method as in Table 6. Specifically, we use 4 crops with the resolution $224 \times 224, 160 \times 160, 128 \times 128, 96 \times 96$ for the student network. The result is shown in Table 11, as we can see the training cost of ReSSL* + Multi-Crops is on par with the SimCLR and BYOL, but our performance is significantly better than all state-of-the-art methods.

Table 11: Working with Multi-crop strategy.

| Method | Epochs | Batch Size | GPU | GPU Memory | (GPU·Time)/Epoch | Acc |
|--------|--------|-----------|-----|-----------|------------------|-----|
| SimCLR | 200 | 4096 | 32 x V100 | 858 G | 3.55 | 66.8 |
| BYOL | 200 | 4096 | 32 x V100 | 863 G | 3.88 | 70.6 |
| SimSiam | 200 | 256 | 8 x V100 | 58 G | 3.51 | 70.0 |
| MoCoV2 | 200 | 256 | 8 x V100 | 40 G | 2.25 | 67.5 |
| ReSSL (Ours) | 200 | 256 | 8 x V100 | 40 G | 2.25 | 68.7 |
| ReSSL* (Ours) | 200 | 256 | 8 x V100 | 42 G | 2.33 | 69.6 |
| ReSSL* + Multi-Crops | 200 | 256 | 8 x V100 | 80 G | 3.62 | **73.8** |