# OpenReview forum: "ReSSL: Relational Self-Supervised Learning with Weak Augmentation"
_NeurIPS.cc/2021/Conference — NeurIPS 2021 Poster_

### Official Review · Reviewer_5Qa3 · 2021-07-15

**Rating:** 7
**Confidence:** 4

**Summary:**

This paper proposes a self-supervised learning technique, ReSSL, which makes (1) similar samples more similar and (2) dissimilar ones be more dissimilar. Since the proposed method does not assume positive and negative samples strictly, it does not suffer from such a class collision problem. Extensive experiments demonstrate the effectiveness of the proposed method from small-scale (e.g., CIFAR) to large-scale (i.e., ImageNet) datasets.


**Limitations And Societal Impact:**

This paper does not address limitations and potential negative societal impacts.

**A suggestion for negative societal impact.**
The line of self-supervised learning works, including this paper, requires a large neural network, a huge training dataset, and a long training time. Hence, it could consume a lot of energy for training, i.e., result in some environmental issues. To solve the problems, we may need an energy-efficient approach. For example, curriculum learning would be one option for improving the efficiency of self-supervised learning.


**Main Review:**

Strengths
- The proposed idea is simple and novel.
- The proposed method demonstrates its effectiveness in both small- and large-scale benchmarks.

Weaknesses & Concerns & Suggestions
1. Few transfer learning experiments.
   - This paper only considers the PASCAL benchmark with linear evaluation for transfer learning scenarios. However, self-supervised learning aims to transfer the learned representations or whole network parameters into various downstream tasks. This is why many SSL methods are often evaluated in various transfer learning scenarios. Could you provide more transfer learning experiments, for example, linear evaluation and fine-tuning in fine-grained classification tasks [2,3], semi-supervised learning [2,3,5], and object detection/segmentation [1,4]?
2. Small number of training epochs
   - The convergence rate and final accuracy highly depend on methods. For example, as reported in [4, Appendix D and Table 4], SimCLR's accuracy is more rapidly increasing than SimSiam in CIFAR-10, but SimCLR's final accuracy (91.1% at epoch 800) is lower than SimSimam (91.8%). Therefore, the reported results in this paper only show that ReSSL has a higher convergence rate (i.e., higher efficiency) than others. I think longer training experiments would further verify ReSSL's effectiveness.
4. Ablation study
   - Student's augmentations
     - I agree that the weak augmentation is important for the teacher network, as shown in Table 3 and Table 9.
     - I wonder why this paper does not conduct a similar empirical study for student's augmentations. Like CLSA, Can ReSSL use stronger augmentations?
   - Although ReSSL achieves higher accuracy than other methods in various settings, it is hard to fully understand why the method succeed. I think more qualitative and quantitative analyses would provide a better fundamental understanding.
     - More t-SNE comparisons with negative-free approaches (e.g., SimSiam) and clustering-based ones (e.g., SwAV) on other datasets (e.g., STL10)
     - Could you provide the changes of the k-NN classification accuracy during pretraining and compare it with other methods, especially clustering-based one? In my opinion, the proposed method is somewhat similar to clustering. Hence, the method might have an advantage when using the k-NN accuracy instead of linear evaluation.
5. Is this method applicable to other domains?
   - SimCLR and MoCo have shown their effectiveness in non-vision domains such as tabular data and natural language. If ReSSL is also widely applicable, it will strengthen this paper's contribution.
6. Some important results are provided in only the supplementary material, not the main manuscript.
   - Multi-crop experiments (Appendix E)
   - Ablation study on the choice of data augmentations (Appendix C)
   - I think this paper does not use its space compactly; for example, Table 5 and Table 6 can be combined. Hence, it would strengthen this paper if the main manuscript contains all the experimental results.
7. Some concurrent works
   - There are concurrent works [6,7,8] that share similar concepts of the proposed method. It might be better if comparisons between them and ReSSL are provided. (since they are concurrent works, I think the information can be placed in the supplementary material).

[1] He et al., Momentum Contrast for Unsupervised Visual Representation Learning, CVPR 2020 \
[2] Chen et al., A Simple Framework for Contrastive Learning of Visual Representations, ICML 2020 \
[3] Grill et al., Bootstrap Your Own Latent: A New Approach to Self-supervised Learning, 2020 \
[4] Chen & He, Exploring Simple Siamese Representation Learning, 2020 \
[5] Caron et al., Unsupervised Learning of Visual Features by Contrasting Cluster Assignments, NeurIPS 2020 \
[6] Dwibedi et al., With a Little Help from My Friends: Nearest-Neighbor Contrastive Learning of Visual Representations, 2021 \
[7] Wang et al., Solving Inefficiency of Self-supervised Representation Learning, 2021 \
[8] Caron et al., Emerging Properties in Self-Supervised Vision Transformers, 2021


**Time Spent Reviewing:**

6

---

> ### Author Response · Authors · 2021-08-10
> **Thank you for your review**
>
> ## Re Q1: few transfer learning experiments.
> Following your suggestions, we have enriched the transfer learning experiments on various downstream tasks. See right down below.
>
> **Fine-grained classification tasks.**
>
> |          | CIFAR10 | CIFAR100 | Food101 | Cars |
> |----------|:----:|:---:|:---:|:---:|
> |  BYOL(1000epochs) | **97.8** | 86.1 | 88.5 | **91.6** |
> |  SimCLR(1000epochs) | 97.7 | 85.9 | 88.2 | 91.3 |
> |  ReSSL(200epochs) | **97.8** | **86.7** | **88.6** | 91.2 |
>
> Following the setting in BYOL and SimCLR, we implement fine-grained classification on Cifar10/100, Food101, and Cars datasets. In detail, we pretrain ReSSL for 200 epochs and adopt 5 crops with the resolution 224×224, 196x196, 160×60, 128×128, 96×96 for the student network. Then we finetune the pretrained model on Cifar10/100, Food101, and Cars datasets; it is clear that our proposed method is comparable with SimCLR and BYOL but with much lesser training cost (200 epochs vs 1000 epochs).
>
> **Semi-supervised learning.**
>
> |          | 1% | 10% |
> |----------|:----:|:---:|
> |  SimCLR(1000epochs) | 48.3 | 65.6 |
> |  BYOL(1000epochs) | 53.2 | 68.8 |
> |  SwAV(800epochs) | 53.9 | 70.2 |
> |  ReSSL(200epochs) | **57.9** | **70.4** |
>
> Following the setting in SimCLR, BYOL and SwAV, we implement semi-supervised learning by fine-tuning the pretrained self-supervised model with 1% and 10% labeled samples on ImageNet. In this experiment, we adopt the same pretraining strategy as we mentioned above. Notably, with just 200 epochs of pretraining, ReSSL outperforms all previous methods. (200 epochs vs 800 / 1000 epochs)
>
> **Object detection.**
>
> | BYOL | SwAV | SimCLR | ReSSL | MoCo v2 | SimSiam |
> |:----:|:----:|:---:|:---:|:---:|:---:|
> | 81.4 | 81.5 | 81.8 | 82.2 | 82.3 | **82.4** |
>
> Results ( VOC07+12 Faster RCNN AP50) are copied from SimSiam paper; all models are optimized for 200 epochs. In this experiment, we only adopt two 224x224 samples for the student network which matches the SimSiam paper's training setting.  As we can see, ReSSL has better performance than SimCLR, BYOL, SwAV and achieved comparable results with previous SOTA methods.
>
> ## Re Q2: small number of training epochs.
> To comprehensively show the effectiveness of ReSSL for longer training, we have tried to run ReSSL on Cifar10/100 for 800 epochs, we provide the result for KNN evaluation for 200/400/600/800 epochs, respectively, and the linear evaluation result for 800 epochs. As we can see, ReSSL is still better than other methods in various conditions.
>
> **Resul for Cifar10**
>
> |          | 200 epochs | 400 epochs | 600 epochs | 800 epochs |  Linear Eval |
> |----------|:----:|:---:|:---:|:---:|:---:|
> |  SimCLR | 86.2 | 88.9 | 89.9 | 90.1 | 91.1 |
> |  SwAV | 86.2 | 87.8 | 88.4 | 89.0 | 90.2 |
> |  SimSiam | 85.8| 88.8 | 90.5 | 90.8 | 91.8 |
> |  ReSSL | 87.1 | 89.1 | 91.4 | 92.5 | **92.5** |
>
> **Resul for Cifar100**
>
> |          | 200 epochs | 400 epochs | 600 epochs | 800 epochs |  Linear Eval |
> |----------|:----:|:---:|:---:|:---:|:---:|
> |  SimCLR | 56.6 | 59.7 | 62.1 | 62.2 | 62.9 |
> |  SwAV | 57.1 | 60.4 | 61.2 | 61.4 | 63.0 |
> |  SimSiam | 56.1 | 63.0 | 64.3 | 64.8 | 67.1 |
> |  ReSSL | 58.3 | 62.8 | 65.6 | 67.9 | **68.6** |
>
>
> ## Re Q3: ablation study
> **Student’s augmentations.**
> The augmentation is a critical point in SSL, so for fair comparison with various baselines, we simply adopt the most commonly used augmentation as SimCLR and MoCo.  Actually, we have also tried the CLSA-style stronger augmentations, in this case ReSSL* achieved 69.4% (ImageNet) which is slightly lower than the standard contrastive augmentation (69.6%) since stronger augmentation generally requires longer training to converge, which is time-consuming and not efficient. But we will further investigate the effect of stronger augmentation besides the common augmentations used in our paper.
>
> **More t-SNE visualization comparisons.**
> We have provided the t-SNE visualization result (stl10)  for SwAV, SimSiam, and ReSSL at https://anonymous.4open.science/r/ReSSL-C3DE/images/stl10-tsne.png .  From this visualization result, we can see clearly that ReSSL has a better linear separability than SwAV and SimSiam. Note that although SwAV is a clustering-based method, it generally requires a large number of prototypes to get better performance. In our experiments, we have tried different numbers of prototypes (10, 30, 50, 100, 300, 500) and we find that the best performance comes from 300 and 500. The same phenomenon was also found in the original SwAV paper (Table11).  This could explain why SwAV has lots of small groups in the visualizations.
>
> **More results of changes of KNN classification accuracy during pretraining.**
> The KNN evaluation result has been provided in the **[Re Q2: Small number of training epochs]**. We agree that ReSSL has some advantages under the KNN evaluation since the gap between KNN and linear evaluation is much smaller than other methods, and ReSSL achieves consistent superiority over baselines.
>
>
> ## Re : Q4: effectiveness on other domains
> ReSSL also works on Tabular data. To verify this, we adopt the tabular dataset Fashion-Mnist, which is regarded as a 784-d vector for each 28*28 image. Then, we use a 12-layer MLP (with BN and ReLU) as our encoder. We also apply a Gaussian noise (mean=0, variance=1) to generate a strong augmented sample and use the clean data as our target sample. In this case, the supervised baseline achieved 90.8% Top-1 accuracy, and ReSSL achieved 85.1% Top-1 accuracy under linear evaluation protocol with no label used in self-supervised pretraining.
>
> Because of the lack of GPU resources, we are not able to test ReSSL on NLP tasks in a limited time. But we believe that ReSSL will still work for NLP tasks theoretically since ReSSL only captures the similarities among examples (e.g. sentences) to fulfil the representation learning. We would like to validate it in the future to further strengthen the contribution.
>
> ## Re Q5: use space more compactly.
> Thank you for your valuable suggestion, we would like to move Appendix E and C to our manuscript, and also merge Table 5 and Table 6 to improve the quality of our paper.
>
> ## Re Q6: Comparing to concurrent works.
> **Comparing with NNCLR [6] (29 Apr 2021) :**  This paper proposes a Nearest-Neighbour based contrastive learning method (NNCLR) where the most similar samples will be regarded as the positive pair. However, the batch size dramatically affects the performance (see NNCLR Table 7) but ReSSL works very well under 256 batch size. On the other hand, NNCLR still defines absolute negative samples, which will still suffer from the class collision problem. Note that ReSSL outperforms NNCLR under semi-supervised setting with just 200 epochs pretraining.
>
> **Comparing with median triplet [7] (18 Apr 2021) :**  This paper reveals an interesting under- clustering and over-clustering problem in contrastive learning and proposes a median triplet loss which defines the rank-k sample as the negative. This is a partial solution to the class collision problem since the most similar samples will not be treated as negatives. However, because of the existence of the clear negative sample, median triplet is still suffers from the class collision problem. ReSSL also has a slightly better performance when working the multi-crop strategy. . If we adopt 5 crops with the resolution 224×224, 196x196, 160×60, 128×128, 96×96 for the student network, (this training setting is only 15% slower than BYOL) ReSSL could achieve 74.7 Top-1 accuracy (vs median triplet 74.1) under linear evaluation protocol.
>
>
>
> **Comparing with DINO [8] (29 Apr 2021) :** This paper proposes a self-distillation based method (DINO) for self-supervised learning. Different from ReSSL, DINO adopts a prototypical layer so the similarity distribution is measured between the instances and the class centers, whereas in ReSSL the similarity distribution is directly computed from different instances which is a more straightforward way. On the other hand, DINO also adopts the “centering” and “sharpening” technique to avoid the model collapsing, but in ReSSL we only use “sharpening” which makes our framework simpler and easier to implement. Since DINO mainly focuses on self-supervised learning for transformer architectures, we are not able to perform quantitative comparisons, but we would like to further investigate the performance of ReSSL on transformers when we have enough GPU resources.
>
> We would like to add more comparisons with these concurrent works in the supplementary materials.
>
> ## Re: Limitations and societal impact.
> We agree that current self-supervised learning framework (including ReSSL) always requires a lot of computational resources which would be the limitation and negative societal impact of our work. We would like to consider your valuable idea (e.g. curriculum learning) to explore a more efficient self-supervised learning algorithm.

---

> > ### Comment · Reviewer_5Qa3 · 2021-08-14
> > **Thank you for the response! Most concerns have been resolved, but one (somewhat minor) concern remains.**
> >
> > Thank you very much for the response. I sincerely appreciate your effort to address my concerns and questions. I believe that the above additional empirical results and discussions would strengthen this work.
> >
> > Most concerns have been resolved, but one (somewhat minor) concern remains: for transfer learning experiments with fine-grained downstream tasks, why did you use only fine-tuning and only a few datasets? Could you test both linear evaluation and fine-tuning with more (3~4?) datasets? Although I already agree with the ReSSL's superiority over baselines, I think such a comprehensive empirical analysis is required, especially on this topic. I will raise my score if the results are still comparable (not necessary to outperform) with the baselines.
> >
> > (Optional) Also, I have one suggestion while this is independent of my score: could you try to use a prediction MLP as BYOL, SimSiam, DINO, MoCo-v3 did? This architecture might be helpful to improve ReSSL's performance.

---

> > > ### Author Response · Authors · 2021-08-18
> > > **Thank you for your prompt response !**
> > >
> > > Thank you for your prompt response; we have conducted more fine-grained downstream tasks on both fine-tuning and linear evaluation settings to improve our comprehensiveness of empirical analysis. The results are shown below.
> > >
> > > ## Fine-grained classification tasks (Fine-Tuning)
> > >
> > > |          | CIFAR10 | CIFAR100 | Food101 | Cars | DTD | Pets | Flowers |
> > > |----------|:----:|:---:|:---:|:---:|:---:|:---:|:---:|
> > > |  BYOL(1000epochs) | **97.8** | 86.1 | 88.5 | **91.6** | 76.2 | 91.7 | **97.0** |
> > > |  SimCLR(1000epochs) | 97.7 | 85.9 | 88.2 | 91.3 | 73.2 | 89.2 | 97.0 |
> > > |  ReSSL(200epochs) | **97.8** | **86.7** | **88.6** | 91.2 | **77.3** | **92.9** | 96.7 |
> > >
> > > In the fine-tuning experiment, we further add DTD, Pets, and Flowers datasets to verify the effectiveness of ReSSL pretraining. As we can see, ReSSL can outperform previous methods on most datasets. Moreover, we would like to note that **we do not perform any extensive hyperparameter search** as in BYOL and SimCLR; most of our experiments are based on the same hyperparameters (e.g., learning rate and weight decay.)
> > >
> > >
> > > ## Fine-grained classification tasks (Linear Evaluation)
> > >
> > > |          | CIFAR10 | CIFAR100 | Food101 | Cars | DTD | Pets | Flowers |
> > > |----------|:----:|:---:|:---:|:---:|:---:|:---:|:---:|
> > > |  BYOL(1000epochs) | 91.3 | 78.4 | **75.3** | **67.8** | **75.5** | **90.4** | **96.1** |
> > > |  SimCLR(1000epochs) | 90.6 | 71.6 | 68.4 | 50.3 | 74.5 | 83.6 | 91.2 |
> > > |  ReSSL(200epochs) | **93.8** | **78.6** | 74.8 | 67.4 | 75.3 | 89.2 | 94.4 |
> > >
> > > Following the linear evaluation experiments in BYOL and SimCLR, we extract the image representations from the average pooling layer and minimize the cross-entropy objective using L-BFGS with l2-regularization. As we can see, ReSSL could outperform the previous methods on the Cifar10/100 dataset. Although BYOL has better performance on the rest of the datasets, the performance gap between ReSSL and BYOL is very narrow, but our method clearly is superior to the baseline SimCLR. Moreover, we only optimize ReSSL for 200 epochs, but the performance is already competitive to that of BYOL and SimCLR for 1000 epochs.
> > >
> > >
> > > ## Working with Predictor (Linear Evaluation)
> > >
> > > |          | CIFAR10 | CIFAR100 |
> > > |----------|:----:|:---:|
> > > |  w/o Predictor | 92.5 | 68.6 |
> > > |  w/ Predictor | **93.0** | **70.5** |
> > >
> > > Thank you so much for your valuable suggestion! To verify the effectiveness of the predictor head, we adopt the same hyperparameters and test this setting on Cifar10/100 for 800 epochs. We surprisingly find that the predictor head can actually further improve the performance as above. So the predictor acts like a plug-and-play component to further enhance the superiority of our ReSSL. We would like to conduct more comprehensive experiments to further analyze the effectiveness of the predictor on various datasets (especially ImageNet). Once we have completed more experiments, we will immediately update them in our supplementary materials.

---

> > > > ### Comment · Reviewer_5Qa3 · 2021-08-18
> > > > **Thank you for the response! I will update my score.**
> > > >
> > > > Thanks for providing this response. I sincerely appreciate again your efforts to address my additional requests. I will raise my score to 7. I believe that all the experiments conducted in this rebuttal period are meaningful and would advance this paper's contribution. I hope that they will be carefully incorporated into the final draft.

---

### Official Review · Reviewer_HMe7 · 2021-07-17

**Rating:** 7
**Confidence:** 4

**Summary:**

This work focuses on self-supervised learning for image classification. Previous work assumes that images and the corresponding augmentation should have similar features (or cluster into the same class) as the objective function to train the model. In contrast, this paper proposes to make the contrastive loss (positive and negative) of two different weak augmentations of a sample to be consistency. Moreover, knowledge distillation is also adopted to sharpen the target distribution for better capturing the most important relationship. Experimental results show that the proposed approach outperforms state-of-the-art approaches, e.g, SimCLR, BYOL, SimSiam, MoCoV2, with less computation requirement.

**Ethical Concerns:**

No.

**Limitations And Societal Impact:**

No. I cannot find the statements of limitations in the supplementary materials, even though this paper states "Please refers to our supplementary materials" in line 402.

**Main Review:**

Generally speaking, this paper is well-motivated and easy to follow. The results show promising improvement on the state-of-the-art methods. Several minor issues are listed below.
1) Some abbreviations require the full name to make them friendly for new readers, e.g., EMA.
2) Some experimental results lack of detailed descriptions. For example, Figure 3 shows the t-sne of MoCo-v2 and the proposed ReSSL. However, it is difficult to tell which one is better. The brown ones still locate cross the green ones. Most importantly, the visualization does not show the reason why the proposed approach is better. It is encouraged to show the ablation visualization, i.e., Full model, Full model without sharpening, full model using strong augmentation.
3) It is desirable to release the codes for reproducing the results.

**Time Spent Reviewing:**

2.5 hours

---

> ### Author Response · Authors · 2021-08-10
> **Thank you for your review**
>
> ## Re Q1: abbreviations.
> Thanks for your suggestion. We will indicate the full name of ALL abbreviations used in our paper to improve the readability. To name a few,
> * EMA ---> exponential moving average
> * SSL  ---> self-supervised learning
> * NCE ---> noise contrastive estimation
>
> ## Re Q2: lack of detailed descriptions and visualization.
> Thank you for pointing out this description issue. We would like to add more details in our final version.
>
> For Figure 3 (t-SNE visualization), a good pretrained model should have a better linear separability for different classes. Although the brown points are still located across the green points, we can see a larger margin between the red points & orange points, turquoise points & purple points, turquoise points & orange points. Since different color represents different classes, this visual margin infers the effectiveness of ReSSL.
>
> Besides, Figure 3 is generated using the Cifar10 training set with 1k t-SNE iterations, we found that 100k t-SNE iterations would be a better option to show the difference between different models, and do visual ablations. Hence, we provide more visualization results at https://anonymous.4open.science/r/ReSSL-C3DE/images/ablation-tsne.png , which includes the visualizations for ReSSL (full model without sharpening), ReSSL (full model with contrastive augmentation), ReSSL (standard full model) and MoCov2.  As we can see, the model will collapse without the sharpening, which matches the result in Table 8 of our supplementary materials. On the other hand, the figure also shows that using contrastive augmentation to generate the target similarity distribution will hurt the linear separability of representations. Finally, it is clear that the standard ReSSL (full model) could result in much better intra-class compactness and inter-class discrepancy than the baseline MoCov2.
>
> We will include these visualization results in our final version.
>
> ## Re Q3: code release.
> We have provided our code and pre-trained model at https://anonymous.4open.science/r/ReSSL-C3DE/README.md
>
> ## Re Limitations and societal impact:
> The major limitations of our work are two-fold.
> 1. Self-supervised learning generally requires much more computational resources than supervised learning.
> 2. Self-supervised learning generally has a poor performance on small networks. Although ReSSL is more friendly to small networks compared with previous methods (Supplementary material Table10), the performance is still much lower than supervised learning.
>
> The negative societal impact would be the environmental issue since self-supervised learning needs lots of energy for training, which will cost a lot of electric power.

---

> > ### Comment · Reviewer_HMe7 · 2021-08-19
> > **Thanks for the reply.**
> >
> > The reply addresses my concern. After rechecking the paper, one quick question about the method is that the memory buffer always replaces the oldest one with the newest one. Is this the best approach? Can we have a better approach for constructing the memory buffer? Thanks.

---

> > > ### Author Response · Authors · 2021-08-19
> > > **Thank you for your response !**
> > >
> > > Thank you for your response! In ReSSL, the memory buffer stores a group of examples to model the pairwise similarity among samples (i.e., relational distribution). Inspired by the same technique in many benchmark self-supervised learning algorithms (MoCo v1/v2, SimCLR v2, CLSA, PCL, MoCHi), we also follow the same rule to construct the memory buffer, i.e., FirstInFirstOut in time order. Actually, this makes sense since the feature(embedding) is updated dynamically; calculating the pairwise similarity with outdated features will be less reliable and meaningful. Thus we expect the features in the memory buffer to be always up to date.
> > >
> > > Of course, this FIFO rule only considers the temporal property of examples in the memory buffer, and is not optimal(best) rigorously. For example, considering some characteristics of examples besides the time order might further benefit the memory buffer. However, what we emphasize in this paper is that with a simple memory buffer, our ReSSL can already have a superior performance by modeling the relation among examples. And we believe a more advanced and sophisticated manner to construct the memory buffer will bring further improvement to our ReSSL, which we would like to leave as future work.

---

### Official Review · Reviewer_Wh5r · 2021-07-17

**Rating:** 6
**Confidence:** 4

**Summary:**

This paper presents a loss function modification on top of the MoCoV2 framework and seem to provide consistent gain over the original MoCoV2 framework. To make it work better, the author also had to change how images are augmented during training.

**Limitations And Societal Impact:**

Adequately addressed.

**Main Review:**

Existing SSL (self-supervised learning) works usually use contrastive / consistency loss (either way it’s instance as class). One long-standing problem is the so-called collision problem where the model, during SSL, learns different instances as different classes while they may come from the same semantic class.

Suppose there is an image x, and x1, x2 are two augmented version of it.
The authors proposed to not directly use contrastive loss on the features of two augmented versions of the same image. Instead, they propose to first draw a similarity distribution for x1/x2 over K number of recent samples. Then use KL-divergence to align these two distributions. In various experiments, this modification to the contrastive loss seems to provide benefits to the original MoCo V2 framework.

Question 1.
Will the proposed method work for fine grained dataset such as CelebA (faces) or Birdsnap (different birds)? I’m interested to know because the foundation of the proposed loss function is:
Given three samples, x,y,z, if x1 is similar to y but different to z, x2 should maintain such relationships.

Suppose x, y, z are three samples from three distinct dog classes. if the x-species dog is a mix breed from y and z. Then say x1 and x2 are two different crop augmentation of the x-dog. Then depending on where the crop happens (e.g., ears, paws, tail), x1 could indeed be similar to y and x2 could be actually more similar to z.

Question 2.
The SWaV number (69.1) in Table 5 seems low and I understand that the authors copied it from citation_10. But when I look at SWaV itself (https://arxiv.org/pdf/2006.09882.pdf), they report in their in Table 3with ResNet50 as 72.7 with 200 epochs. May I ask why is there such a gap?

**Time Spent Reviewing:**

1.5 hours

---

> ### Author Response · Authors · 2021-08-10
> **Thank you for your review**
>
> ## Re Q1: effectiveness on fine-grained dataset.
> Our ReSSL still works for fine-grained dataset.  We understand the reviewer might worry about the cropping area issue. However, for augmentations, we set the lower bound of crop ratio to 0.2 as we have mentioned in section 4.2 (comparing to 0.08 for supervised learning generally), so the augmented images are unlikely to just contain a very small area (e.g. only ​​ears, paws, tail). Moreover, we have verified that weaker augmentations usually induce more accurate relationships (similarities) among examples (Table 3 and Figure 3), so the generated target distribution is more reliable.
>
> To verify the effectiveness of ReSSL on the fine-grained dataset, we performed an attribute classification task on the CelebA dataset. Specifically, we adopt resnet18 as our backbone encoder and resize all images to 64x64 resolution for fast experiments. The supervised baseline achieved 90.8% accuracy in this setting, and ReSSL gets 87.7% under the linear evaluation protocol, demonstrating that ReSSL can also work on the fine-grained datasets.
>
> ##  Re Q2: gap of SwAV number in Table 5.
> The result of SwAV (72.7%) in the original paper adopts the multi-crops strategy (two 224x224 and six 96x96 images) during training.  The reproduced result from citation_10 does not use additional crops (only two 224x224 images are adopted).
>
> The multi-crops strategy can generally improve the linear evaluation accuracy but will result in a much longer training time. We believe that a fair experiment should compare the methods under the same training cost, so we also report the training cost for different methods in Table 6. Note that SwAV with two 224x224 images should have a similar training cost with SimCLR.
>
> Moreover, we also report the result for ReSSL+multi-crops in our supplementary materials (Table 11); in this case, ReSSL achieved 73.8%, the training cost is similar to the 2x backprop methods but the performance is much higher.

---

> > ### Comment · Reviewer_Wh5r · 2021-08-25
> > **thanks for your response**
> >
> > No other questions

---

### Official Review · Reviewer_k9Fo · 2021-07-18

**Rating:** 6
**Confidence:** 3

**Summary:**

This paper proposes a new approach to self-supervised learning of representations, wherein similar to previous works, the objective function relates to two different views–as in prior works, augmentations here–of the same data point.  However, rather than encouraging the respective two embeddings to be similar, the approach here encourages the distribution of similarities to other data points of the two embeddings to be similar, along with a momentum update approach.  On a thorough set of empirical evaluations, the proposed approach performs strongly.

**Main Review:**

Overall, the proposed approach seems like an intuitive modification to recent self-supervised learning approaches, which performs well on a thorough range of empirical evaluations relative to recently proposed approaches, and therefore should be interesting to attendees and readers of this conference.  Some specific points:
- The proposed approach has a number of "tricks" (like the momentum approach, distribution sharpening, particular type of weak augmentation), however these appear to be thoroughly ablated in the empirical results section
- The idea of applying some objective over the matrix of pairwise distances between a data point and others in the dataset is not itself new... ideally this paper would have compared to e.g. kernel-based approaches and mapped connections there
- The writing is somewhat unclear and has grammatical issues throughout, especially making some of the upfront exposition parts of the paper a little hard to parse.

**Time Spent Reviewing:**

3

---

> ### Author Response · Authors · 2021-08-10
> **Thank you for your review**
>
> ## Re Q1: a number of “tricks”.
> Our ReSSL proposes a novel self-supervised learning paradigm by investigating the similarities among examples when pulling closer the different augmentations. To fulfil the pulling augmentations, using a momentum encoder is a common way which is also widely used in MoCo and BYOL. In this way, for better representation learning, two key components of ReSSL is first to form a more reliable similarity distribution and then to concentrate more on the similar examples, which can be exactly realized by the proposed simple “weak augmentation” and “sharpened distribution”. Therefore, both techniques serve as critical parts in ReSSL and matches the logical idea well instead of just being “two tricks”. Of course, since we emphasize the effect of them, we did implement comprehensive and solid ablation studies about them. Empirical results show their effectiveness and validate the correctness of the idea of ReSSL.
>
> ## Re Q2: connections to kernel-based approaches.
> Thanks for pointing out this. Actually, our method can be also regarded as a special kernel-based method. Note that the core of ReSSL is to capture the similarity relationships among different examples instead of defining absolute positive and negative pairs as other baseline methods. So we simply adopt the exponential of the pairwise inner product as similarity metric (Eq3 and Eq4), which is exactly the Gaussian kernel, and use SoftMax normalization to form the distribution to pull the augmentations. This practice can be regarded as a normalized Gaussian method, and is similarly used in SEED (line 165). However, what we emphasize in ReSSL is not the type of kernel we use, but the idea of investigating similarity among examples. And we can see from the empirical results, a simple kernel (similarity) used in ReSSL already has significant superiority over many SSL baselines. Of course, we believe a more advanced or sophisticated kernel might bring further improvement for our ReSSL, which we will investigate in our future work.
>
> ## Re Q3: writing issue.
> Thank you for pointing out the writing issue, we will proofread carefully and make our paper easier to understand in the final version.

---

> > ### Comment · Reviewer_k9Fo · 2021-09-03
> > **Response to author response**
> >
> > First of all: thank you for your detailed response to my review, and apologies for the delay in responding in turn on my end!
> >
> > Thank you for the clarification re: ablations and the connection to kernel-based approaches!  I will leave my review as is at this time, which is inclined towards acceptance; I greatly appreciate the reviewer's responses and additions.

---

### Decision · Program_Chairs · 2021-09-27

**Decision:**

Accept (Poster)

**Comment:**

This paper proposes a more nuanced take on contrastive self-supervised learning (SSL). Rather than frame the objective of SSL as a binary similar/not similar target, the authors propose to use multiple sampled augmentations as a similarity distribution. The goal of SSL becomes to embed the training data in a way that agrees with the distribution. The result can be viewed either as a relaxed form of contrastive SSL, in which multiple views of the same instance only need to be similar, not the same, or---as one reviewer pointed out---a kernel-based approach that operates on the similarities among instances.

While the idea is intuitively appealing, the authors show that simply plugging it into existing SSL setups gives very poor performance, so the authors provide a recipe for this alternative approach. In particular, only weak augmentations (as opposed to changing the instances heavily) leads to better results, in contrast with existing contrastive learning approaches where the augmentation must be strong to make the pretext task challenging. Experimental results show that the learned representations lead to significant gains on object classification tasks.

The reviewers generally agree that this paper is well written and makes a significant contribution. During the discussion phase, the authors introduced several follow up experiments, which the reviewers encourage them to include in the final version.